# Dataset Bias Mitigation in Multiple-Choice Visual Question Answering and Beyond

**Zhecan Wang[1], Long Chen[2][*], Haoxuan You[1][*], Keyang Xu[1], Yicheng He[1]**
**Wenhao Li[1], Noel Codella[3], Kai-Wei Chang[4], Shih-Fu Chang[1]**

[1] Columbia University   [2] HKUST   [3] Microsoft Research   [4] University of California, Los Angeles

{zw2627, hy2612, yh3330, wl2750, sc250}@columbia.edu

longchen@ust.hk, xky0714@gmail.com, ncodella@microsoft.com, kwchang@cs.ucla.edu

## Abstract

Vision-language (VL) understanding tasks evaluate models' comprehension of complex visual scenes through multiple-choice questions. However, we have identified two dataset biases that models can exploit as shortcuts to resolve various VL tasks correctly without proper understanding. The first type of dataset bias is *Unbalanced Matching* bias, where the correct answer overlaps the question and image more than the incorrect answers. The second type of dataset bias is *Distractor Similarity* bias, where incorrect answers are overly dissimilar to the correct answer but significantly similar to other incorrect answers within the same sample. To address these dataset biases, we first propose Adversarial Data Synthesis (ADS) to generate synthetic training and debiased evaluation data. We then introduce Intra-sample Counterfactual Training (ICT) to assist models in utilizing the synthesized training data, particularly the counterfactual data, via focusing on intra-sample differentiation. Extensive experiments demonstrate the effectiveness of ADS and ICT in consistently improving model performance across different benchmarks, even in domain-shifted scenarios.

## 1 Introduction

Visual Question Answering (VQA) is a challenging vision-language task that requires reasoning with integrated information from visual and text modalities (Antol et al., 2015; Zellers et al., 2019; Lei et al., 2020; Ren et al., 2015; Lu et al., 2021; Tapaswi et al., 2016; Schwenk et al., 2022). VQA benchmarks (Zellers et al., 2019; Lei et al., 2020; Tapaswi et al., 2016; Lei et al., 2018, 2019) present complex scenarios with multiple entities in a multiple-choice question format, where the model selects the correct answer from multiple long, context-dependent candidate answers.

Previous studies have examined dataset bias in VQA benchmarks with short-phrase answers

---
[*] Equal Contribution

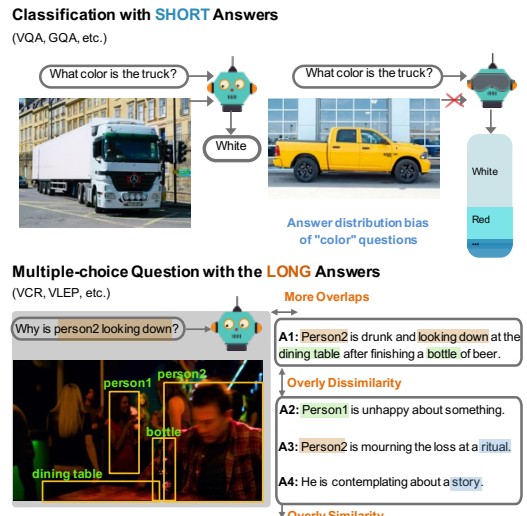

Figure 1: (a): Prior studies examined dataset bias in the distribution of short-phrase answers (*e.g.*, "white" is often the answer when asking about color). (b): Our work investigates the biases in VQA with long answer choices, where the correct answer has more n-grams overlapped with the image and question (orange and green). Meanwhile, the incorrect answers contain more irrelevant n-grams to the scene (blue).

(VQA-Short) (Chen et al., 2020a; Dancette et al., 2021; Gokhale et al., 2020; Gupta et al., 2022; Niu et al., 2021; Ramakrishnan et al., 2018). These benchmarks typically consist of simple questions that can be answered using one or two words, focusing primarily on visual perception. However, as the demand for complex reasoning capabilities has increased, VLU benchmarks incorporating rich text annotation as contextual information have gained popularity, leaning towards the annotation of long answers. In this paper, we uncover dataset biases in **VQA-Long**, which employs long answer formats. We contend that these biases pose greater challenges for mitigation and have a significant impact on the training and evaluation process of supervised models. Specifically, we identify two prominent types of biases in VQA-Long. The first is Unbalanced Matching (UM) bias, characterized by an uneven distribution of n-gram matches be-

tween the answer choices and premise (*i.e.*, image and question). The correct answer often exhibits a higher n-gram overlap with the question or mentions more objects in the image than distracting options, which frequently contain unrelated N-grams. The second type, termed Distractor Similarity (DS) bias, occurs when the model can identify the correct answer without considering the question and image. This bias arises when the correct answer distinctly differs from the distractors, which are highly similar amongst themselves.

The two biases we have identified are also not limited to VQA-Long; they are also present in other VLU benchmarks, including SNLI-VE (Xie et al., 2019) and VLEP (Lei et al., 2020). Capitalizing on these biases, we design a simple algorithm based on heuristic rules without any training. Surprisingly, it yields high-performance comparable to supervised models: 66.29% Q2A accuracy on VCR, a long-form VQA problem on visual commonsense reasoning, 69.77% accuracy on SNLI-VE, and 48.85% on VLEP. These results raise questions about if the existing models truly comprehend the context or rely on shortcuts to answer questions.

Different from the biases identified in VQA-short, the dataset biases we identified in these text-rich datasets are significantly harder to remove. They are affected by several reasons, cross-modal correlations, open-ended text generation, and heavy reliance on human artifacts during annotation. These biases in VQA-Long with rich context are more likely to be text-dependent, causing models to under-utilize visual information and potentially develop false visual dependencies.

In terms of mitigating dataset biases, prior data synthesis approaches (Chen et al., 2020a; Dancette et al., 2021; Gokhale et al., 2020; Gupta et al., 2022; Niu et al., 2021; Ramakrishnan et al., 2018) have demonstrated their effectiveness for VQA-Short; however, they are not suitable for VQA-Long. Additionally, some well-known methods (Chen et al., 2020a; Liang et al., 2020) disrupt the data distribution through superficial text masking or image occlusions (see Figure 2). To overcome these limitations, we propose a novel Adversarial Data Synthesis (ADS) method to mitigate biases in VQA-Long, addressing the under-utilization of visual information and incorrect visual dependency in models. ADS generates synthetic factual and counterfactual text data using ADS-T and synthesizes images using ADS-I. Specifically, ADS-T generates long

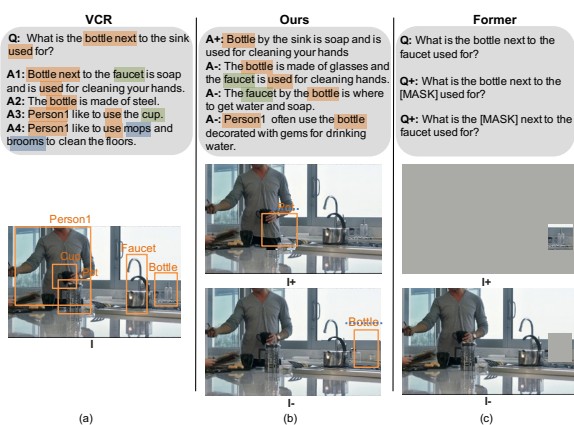

Figure 2: A comparison of image-text data. (a) lists the original VCR question, answer choices, and image data. (b) lists our synthesized factual (I+, A+) and counterfactual (I-, A-) image-text data. (c) shows an example of modifying a sample of VCR data using the former solutions (Chen et al., 2020a; Liang et al., 2020).

sentence answers and distractors directly, while ADS-I generates images that closely resemble real images with minimal disturbance to the data.

Furthermore, previous debiasing methods directly incorporate the synthesized counterfactual questions and images into the models' training input (Figure 2c). However, these methods are not applicable to VQA-Long, as it requires an exact-matched ground truth long answer and distinct distracting options for constructing a multiple-choice question. To address this limitation, we introduce Intra-sample Contrastive Training (ICT), which employs a loss function to promote the models' focus on intra-sample disparities among the synthesized factual, and counterfactual images. This approach guides models to learn the appropriate visual dependency pertaining to the query.

Through extensive comparisons over VCR, SNLI-VE, VLEP, and even VQA-Short datasets, we empirically demonstrate the effectiveness of our methods in improving model performance under both standard evaluation and domain-shifted scenarios. To further assess models' robustness against dataset bias, we create domain-shifted evaluation benchmarks based on VCR using ADS. These benchmarks are validated by human annotators to guarantee high quality. While our analysis and experiments primarily focus on VCR, SNLI-VE, and VLEP, it is important to note that these dataset biases frequently appear in other VLU tasks with rich contextual information.

In summary, our contributions are three-fold:

- We conduct the first comprehensive study on

data biases in VQA-Long, uncovering two prevalent biases in VLU benchmarks.

• We propose a data synthesis method (ADS) and a training strategy (ICT) to address these biases. ADS generates factual and counterfactual image-text data to mitigate biases, while ICT aids models in utilizing this synthesized data during training.

• We introduce evaluation benchmarks to evaluate the robustness of VL models against dataset biases, establishing a more fair comparison for existing and future models.

## 2 Related Work

**Biases in VL Benchmarks.** Previous studies have predominantly focused on biases in VQA-Short (Li et al., 2018; Hudson and Manning, 2019; Lu et al., 2021; Ren et al., 2015; Johnson et al., 2017). These benchmarks lack sample-specific candidate options, resulting in VL models being supervised to classify image-question pairs using shared class labels consisting of short answers. In these datasets, researchers have pointed out that models often downplay visual information and instead focus on learning biases in the text (Agrawal et al., 2018; Dancette et al., 2021; Zhang et al., 2016; Manjunatha et al., 2019). This is exemplified by models directly learning the shallow mapping between prior question words and shared class labels in the absence of sample-specific contextualized candidate options. Consequently, models develop false visual dependency (Cao et al., 2020; Wang et al., 2022b) as they may succeed in resolving VQA tasks (Selvaraju et al., 2016; Chen et al., 2020a; Gupta et al., 2022) utilizing irrelevant visual cues.

However, the biases previously examined may not apply to VQA-Long due to its unique characteristics, such as the absence of shared class labels, the presence of sample-specific candidate options, and the diverse question types. Consequently, it is challenging for models to conclude the question types and determine the most popular answer choices for each question type. Given its difficulty and complex visual scenes, VQA-Long has gained popularity in recent years (Zellers et al., 2019; Tapaswi et al., 2016; Schwenk et al., 2022; Zhu et al., 2016; Li et al., 2020), but bias analysis in this specific context has not been explored extensively. While a recent study (Ye and Kovashka, 2021) briefly addressed bias issues in VCR, it only focused on the exact matching of pronoun words. In contrast, our research delves into a comprehensive analysis of

biases across all textual components, cross-modal correlations in the input, and the process of generating distractors. We identify more general bias problems and demonstrate their prevalence.

**Debiasing Methods.** Various approaches were proposed to counter biases but only focus on VQA-Short. They can be categorized into two directions, training strategies and Data Synthesis (DS), and all suffer from various constraints. For instance, training strategies like (Gupta et al., 2022; Niu et al., 2021) and DS solutions, (Ray et al., 2019; Selvaraju et al., 2020; Ribeiro et al., 2019; Wang et al., 2022d) only focus on a single modality. Debiased training like (Wang et al., 2022b; Niu and Zhang, 2021; Zhang et al., 2021b) require constraints of either a specific model structure or doubling the models' complexity. Other methods (Chen et al., 2020a; Liang et al., 2020) apply occlusion boxes or maskings on images or questions and thus drastically disturb data distribution, leading to nonsensical synthesized answers. Gokhale et al. (2020) tries to improve the synthesized image and text quality but is limited to two specific question types. Most importantly, all of them cannot generalize to VQA-Long. Only a few works, (Wang et al., 2022b,d; Ye and Kovashka, 2021), are related to VQA-Long but still fail to identify specific bias issues.

## 3 Bias Analysis in VQA-Long

In this section, we identify and analyze two distinct types of biases that commonly occur in VQA-Long and other VLU benchmarks.

### 3.1 Unbalanced-Matching Dataset Bias

Inspired by (Ye and Kovashka, 2021), we conducted a comprehensive analysis of matching n-grams within candidate options against the text premise (question), $t$, and visual premise (image)[1], $v$. We calculate the percentage of samples, $C_c^p$ ($C_d^p$) as following, where correct (incorrect) answers $a^c$ ($a^d$) have more matched n-grams ($n \leq 3$) against the premise information $p \in \{v, t\}$ than the other:

$O(a, p) = \#$ matched n-grams between $a$ and $p$

$C_c^p = \frac{1}{N} \sum_{i=1}^{N} \mathbf{1}\{O(a_i^c, p_i) > \max_{a_i^d \in A_i - a_i^c}(O(a_i^d, p_i))\},$

$C_d^p = \frac{1}{N} \sum_{i=1}^{N} \mathbf{1}\{\max_{a_i^d \in A_i - a_i^c}(O(a_i^d, p_i)) > O(a_i^c, p_i)\},$

where for each sample $i$, $A_i$ represents all the paired candidate options for sample $i$, $a_i^c$ is the correct answer, $a_i^d$ is one of the three distractors

---

[1]For matching n-grams against the visual premise, we extract object labels from images.

(incorrect answers), and $p_i$ is the premise (either image or question). $O(a, p)$ is the count of matched n-grams, and N is the total number of samples.

Our analysis reveals that the *correct answer often has the highest number of matched n-grams against the question and image among candidate answer options*. Specifically, for the Q2A task[2] in VCR, $C_c^t$ can be as high as 66.29%, which is much higher than the percentage of distractors, $C_d^t$, at 29.16%. When using image as premise, $C_c^v$ is 42.75% , which is also higher than $C_d^v$, at 40.23%. This unbalance also persists in other VLU benchmarks. For example, $C_c^t$ is 48.85% (higher than $C_d^t$, 36.19%) in VLEP and 69.77% (higher than $C_d^t$, 45.40%) in SNLI-VE. Besides containing fewer n-gram overlaps against the premise, we observed that distractors even contain more irrelevant n-grams to the given context (Details in A.8), as in Figure 5.

## 3.2 Distractor Similarity Dataset Bias

Many benchmarks (Zellers et al., 2019; Williams et al., 2022; Lei et al., 2020; Li et al., 2020; Liu et al., 2020) rely on automated Adversarial Matching (AM) for generating distractors, aiming to minimize costs. AM generates distractors by reusing answers from other questions. It selects answers related to the given question while dissimilar to the correct answer. However, prior works tend to overly emphasize the dissimilarity to the correct answer and thus irrelevance to the context using stylistic models (Details in A.7). Additionally, a significant issue arises from generating distractors without considering visual information in AM (Lei et al., 2020; Zellers et al., 2019; Li et al., 2020). Surprisingly, even in manual annotation settings (Xie et al., 2019; Do et al., 2020; Kayser et al., 2021), annotators are tasked with generating distracting candidates (distractors) without access to the image, forcing them to imagine and create excessively dissimilar distractors to avoid ambiguity. Additionally, insufficient premise can also cause limited diversity of generated distractors. In contrast, correct answers are consistently generated with visual and textual information, ensuring accuracy. Consequently, *the dissimilarity between correct and incorrect answers becomes exaggerated* due to the different premise information used for their generation.

---

[2]Results about QA2R and Q2AR tasks are in A.6.

## 4 Biases Mitigation in VQA-Long

This section introduces (1) Adversarial Data Synthesis (ADS) to synthesize factual and counterfactual data; (2) Intra-sample Counterfactual Training (ICT) method to exploit the data.

### 4.1 Adversarial Data Synthesis

ADS has two components, ADS-Text (ADS-T) for generating less biased answers and ADS-Image (ADS-I) for images.

#### 4.1.1 ADS-T

ADS-T generates synthesized options for a sample, $A+$ and $A-$, to alleviate the dataset biases.

**Multimodal Distractor Generation.** To improve distractors' diversity and relevance to the given context and correct answers, we incorporate visual premise information to improve Adversarial Matching (AM).

For given dataset examples, $\{(p_i, a_i)\}_{i=1}^K$, $p_i$ represents the premise (visual or textual), $a_i$ denotes the answer and $K$ is the total number of samples. Following AM, we utilize the first term in Eq. (1) to measure the relevance of the candidate answer, $a_j$ from other questions, against the premise and the second term to measure the similarity between $a_i$ and $a_j$. Both $S_{\text{t-rel}}$ and $S_{\text{sim}}$ are approximated by stylish models[5]. Further, for every example, $(p_i, a_i)$, we can obtain a distractor by performing maximum-weight bipartite matching on a weight matrix $\mathbf{W} \in \mathbb{R}^{N \times N}$, given by:

$$\mathbf{W}_{i,j} = \log\left(S_{\text{t-rel}}\left(\boldsymbol{p}_i, \boldsymbol{a}_j\right)\right) + \lambda \log\left(1 - S_{sim}\left(\boldsymbol{a}_i, \boldsymbol{a}_j\right)\right),$$
(1)

where $\lambda$ is a hyperparameter. Different from previous works that only include text as the premise, we bring in visual object regions, $v_i^k$, to enrich the reference, where $k \in [0, \mathbb{D}]$ and $\mathbb{D}$ is the total number of object regions extracted from the image $I_i$. We further employ a pre-trained CLIP (Radford et al., 2021) to measure the visual relevance of candidate answers against all of the object regions and only reserve the maximum score, $\max\left(S_{\text{vrel}}\left(\boldsymbol{v}_i, \boldsymbol{a}_j\right)\right)$. Thus, the first term in Eq. (1) will be substituted by: $\alpha \log\left(S_{\text{t-rel}}\left(\boldsymbol{q}_i, \boldsymbol{a}_j\right) + \max\left(S_{\text{v-rel}}\left(\boldsymbol{v}_i, \boldsymbol{a}_j\right)\right)\right)$[5], where $\alpha$ is a hyperparameter.

**Distractor Refinement.** Despite the improvement of our Multimodal Distractor Generation, distractors generated solely based on a few noisy scores or heuristic rules still fall short of perfection. To address this, we conducted in-house annotation of over 100 samples to explore how humans can

further refine answer candidates from Multimodal Distractor Generation. We established specific criteria for quality distractors and hired experienced annotators for iterative refinement (Details in A.3).

Recognizing the improved quality through human refinement, we leverage the largely pre-trained ChatGPT (OpenAI, 2023) to mimic the human refinement process. With rich context and 5 human-annotated examples as input, the model can generalize for large-scale annotations[5].

### 4.1.2 ADS-I

With UM and DS biases appearing to be dominant in text information, existing VL models are encouraged to learn shortcuts in text and under-utilize the visual information leading to false visual dependency, as in Figure 6a (Cao et al., 2020; Wang et al., 2022c; Dancette et al., 2021; Chen et al., 2020a). To assist models in mitigating UM and DS biases, we design ADS-I to synthesize positive images **I+** and negative images **I-** to assist models' training and emphasize the correct question-related visual dependency. We introduce ADS-I in three components: region selection, coarse-to-fine region removal, and finetuning region remover.

**Region Selection.** To generate the synthetic factual images, denoted as **I+**, ADS-I identifies and eliminates irrelevant visual regions referenced to the question-answer pair. Conversely, relevant image regions are removed to create **I-**. Following the procedures outlined in previous work (Chen et al., 2020a), we determine the relevance of image regions based on exact matching and soft scoring of the question-answer pair[5].

**Coarse-to-Fine Region Removal.** Previous studies (Chen et al., 2020a; Liang et al., 2020; Ye and Kovashka, 2021) introduce occlusion boxes or masking tokens, leading to disruptive changes in data distribution and resulting in excessive artifacts, as illustrated in Figure 2(c). In contrast, we propose an innovative approach that transforms the data synthesis task into a visual inpainting task to generate photorealistic synthesized images. To achieve this, we leverage the SPL visual inpainting model (Zhang et al., 2021a) as the base model, comprising two components: $\mathsf{SPL}_p$, pretrained on a large generalized dataset (Zhang et al., 2021a), and $\mathsf{SPL}_f$, finetuned on the specific downstream dataset. Our objective is to implement an effective coarse-to-refine framework utilizing both models to remove image regions iteratively.

To create **I+** by removing a selected irrelevant object, we apply a masking technique using the minimum circumscribed rectangles around its region. The masked image is then fed into the two SPL models for inference. Similarly, for generating **I-**, we employ similar maskings over the relevant regions and infer to remove the corresponding objects. The reconstructed images from the two SPL models can be observed in Figure 2(b). To ensure fine-grained images and avoid superficial artifacts as addressed in prior works (Chen et al., 2020a; Liang et al., 2020; Gokhale et al., 2020), we adopt a coarse-to-refine framework. After the first stage of inpainting with $\mathsf{SPL}_p$, we employ a tri-pass autoregression strategy during the inference with $\mathsf{SPL}_f$. The masked input is iteratively passed through $\mathsf{SPL}_f$ three times in an autoregressive manner, with smaller maskings applied in each subsequent pass. (*c.f.*, Figure 3(a))

**Finetuning Region Remover.** When finetuning $\mathsf{SPL}_f$, we first identify two kinds of visual regions: 1) regions within objects with no other objects on top; 2) regions in the background with no objects on top. Then, inside either kind of these regions, we create rectangular maskings of various sizes. During finetuning, we apply those maskings over input images and supervise $\mathsf{SPL}_f$ to reconstruct or inpaint the masked regions based on their neighboring pixels and patterns (Details in A.4).

## 4.2 Intra-sample Counterfactual Training

We propose ICT to enhance models' training with counterfactual data to facilitate models to focus on intra-sample differentiation, which it is essential in VQA-Long and other VLU tasks with rich context.

### 4.2.1 XE Training

It is common to model VQA tasks as maximizing a probability of answer label selection conditioning on the given image and question, $\hat{P}(\boldsymbol{a} \mid I, Q)$. Thence, models are often supervised to this probability distribution with cross-entropy (XE) loss:

$$L_{\mathrm{XE}} = -\sum_i^K y^i \log\left(\sigma\left(\hat{P}(\boldsymbol{a} \mid I, Q)\right)\right), \quad (2)$$

where $y^i$ is the ground truth label, and $\sigma$ is the softmax function. With XE loss, we can directly bring $I+$, $A+$, and $A-$ into training along with the original data.

### 4.2.2 Answer-focused ICT

Unlike VQA-Short, which focuses on global inter-sample differentiation, VQA-Long tasks emphasize local intra-sample differentiation. The task

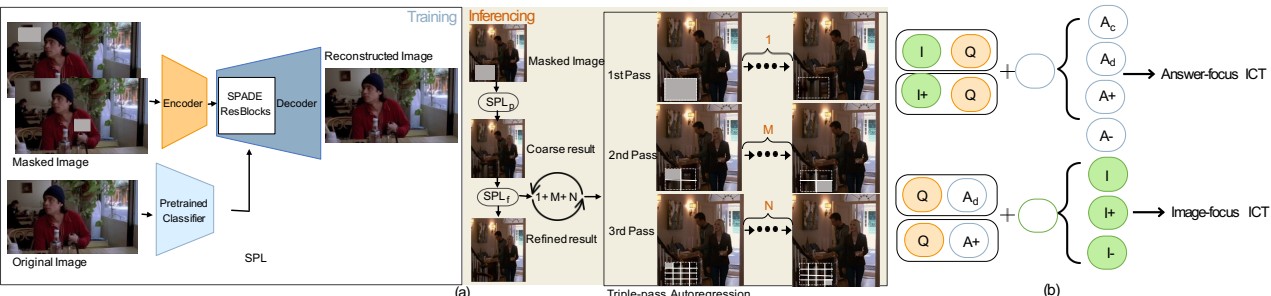

Figure 3: (a) Diagram of Coarse-to-Fine Region Removal. The left part illustrates the training of SPL$_f$ and the right part showcases the coarse-to-fine inferencing process. Within the triple-pass autoregression strategy, the 1st pass includes a one-time inpainting step with the whole region masked, the 2nd pass includes M inpainting steps with smaller masking over regions from the top left to bottom right iteratively, and the 3rd pass includes N steps with further smaller masked regions. (b) Diagram of all the combinations of (+/-I, Q, +/-A) pairs utilized in training. The pairs from the top block are utilized in QA classification training.

focuses on mapping the given premise information to the correct answer among sample-specific candidate options. For instance, in VCR, models learn the local $IQ \rightarrow A$ mapping, each $(i_i, q_i)$ pair against four specific answers, $(a_{i1}, ..., a_{i4})$; In SNLI-VE, the mapping exists between each image and specific candidate hypotheses. Unfortunately, the former methods do not address situations with sample-specific counterfactual candidate options. To incorporate them in VQA-Long tasks, we propose to use InfoNCE (Van den Oord et al., 2018) to measure the intra-sample contrastive loss, $\mathcal{L}_{A-ICT}$ between each $(i_i, q_i)$ against $(a_i, a_i+, a_i-)$:

$$-\log \frac{\exp\left(\Phi\left(\boldsymbol{z}, \boldsymbol{z}_p\right)/\tau\right)}{\exp\left(\Phi\left(\boldsymbol{z}, \boldsymbol{z}_p\right)/\tau\right) + \exp\left(\Phi\left(\boldsymbol{z}, \boldsymbol{z}_n\right)/\tau\right)}, \quad (3)$$

where $\Phi$ measures the cosine distance, $\tau$ is a hyperparameter temperature, $\boldsymbol{z}_p$ is the [CLS] token feature for a positive pair, $(I, Q, A)$ or $(I, Q, A+)$, and $\boldsymbol{z}_n$ is for a negative pair, is $(I, Q, A-)$.

Synthesized answer candidates can have more balanced matched n-grams and more diverse distractor distributions, requiring a stronger model capacity to distinguish (Figure 3(b)). Therefore, Answer-focus ICT can encourage models to focus on this challenging intra-sample differentiation.

### 4.2.3 Image-focused ICT

To be aware, the direct augmentation of counterfactual images, **I**- in training through the aforementioned two training losses still remains unclear, as we cannot find the paired answer choices for **I**-. Previous approaches in VQA v2 (Goyal et al., 2017) require generating new pseudo or soft answer labels (Chen et al., 2020a; Gokhale et al., 2020) for **I**-. However, they are not feasible for VQA-Long tasks such as VCR, which require sample-specific sentence answer options.

We address this issue by transforming the $IQ \rightarrow A$ mapping problem into a $QA \rightarrow I$ mapping problem, which we further narrow to the intra-sample pairing between each $(q_i, a_i)$ pair and $(i_i, i_i+, i_i-)$, similarly utilizing Eq. (3). Existing VL models often underuse visual information, leading to poor visual explainability. By contrasting sample-specific $(i_i, i_i+, i_i-)$, we highlight the significance of relevant visual regions to the question-answer pair. This approach, exemplified in Figure 2, promotes recognition of relevant entities, like the "bottle", and fosters the learning of correct visual dependencies linked to the question-answer pair.

Finally, after combing $L_{A-ICT}$ and $L_{I-ICT}$, the overall objective is:

$$L = \delta_1 L_{XE} + \delta_2\left(L_{A-ICT} + L_{I-ICT}\right), \quad (4)$$

where $\delta_1$ and $\delta_2$ are hyperparameter weights.

## 5 Experiment

**Base models.** Since ADS-ICT is generalizable and can be applied to various models, we evaluated it on several backbones: UNITER$_L$ (Chen et al., 2020b), VL-BERT$_L$ (Su et al., 2019), VILLA$_L$ (Gan et al., 2020) and LMH (Clark et al., 2019).

**Datasets.** We conduct bias analysis and validation experiments over **VCR** (Zellers et al., 2019), **SNLI-VE** (Xie et al., 2019)[3]. Our identified bias problems also apply to other VLU benchmarks with long candidate options, and our methods can even generalize to VQA-Short datasets like VQA v2 and VQA CP2 v2.

### 5.1 Bias Verification

This analysis verifies how the aforementioned two biases affect models' learning and to what extent existing models can take the shortcut for advantage.

---

[3]Results over VLEP (Lei et al., 2020) are provided in the appendix.

| Models | VCR | | | SNLI-VE | |
|---|---|---|---|---|---|
| | Val. | Fair | Adv. | Val | Test |
| Heuristics-Only | 66.29 | 48.70 | 43.93 | 69.77 | 69.30 |
| VL-BERT | 75.51 | 72.84 | 70.46 | 74.66 | 74.71 |
| +Answer Masking | 75.89 | 73.48 | 71.59 | 75.09 | 75.21 |
| +Question Masking | 75.67 | 73.14 | 71.10 | | |
| **+ADS-T** | **76.23** | **73.69** | **72.18** | **75.52** | **75.60** |
| +Occlusion Box ⋆ | 76.07 | 73.67 | 71.83 | 75.24 | 75.52 |
| **+ADS-I ⋆** | **76.88** | **75.03** | **73.00** | **75.90** | **75.96** |
| +CSS | 75.94 | 73.85 | 71.35 | | |
| +CC | 76.04 | 74.29 | 71.72 | | |
| **+ADS-ICT** | **77.33** | **76.12** | **73.72** | **76.27** | **76.33** |
| UNITER | 76.72 | 74.99 | 72.48 | 79.02 | 79.19 |
| **+ADS-ICT** | **78.23** | **77.36** | **74.74** | **80.14** | **80.23** |
| VILLA | 78.28 | 76.67 | 74.05 | 79.64 | 79.32 |
| **+ADS-ICT** | **78.98** | **77.93** | **75.38** | **80.87** | **80.28** |

Table 1: Accuracies (%) on VCR (Q2A) and SNLI-VE based on our re-implementation. "Val" indicates the validation set of VCR. ⋆ indicates training with ICT to utilize counterfactual images. The results of Heuristics-Only are obtained by taking the best performance from a mix of heuristic rules utilizing the two biases, *e.g.*, the method always selects the option with the most matching n-grams.

| A+ | A- | A-ICT | I+ | I- | I-ICT | $VCR_{Std}$ | $VCR_{Fair}$ |
|---|---|---|---|---|---|---|---|
| | | | | | | 75.51 | 72.84 |
| ✓ | | | | | | 75.80 | 73.05 |
| ✓ | ✓ | | | | | 76.23 | 73.69 |
| ✓ | ✓ | ✓ | | | | 76.85 | 74.46 |
| ✓ | ✓ | ✓ | ✓ | | | 76.93 | 75.08 |
| ✓ | ✓ | ✓ | ✓ | ✓ | ✓ | **77.33** | **76.12** |

Table 2: Ablation study on $\text{UNITER}_L$. A-ICT is answer-focused ICT, and I-ICT is image-focused ICT.

| Models | VQA-CP v2 test | | | |
|---|---|---|---|---|
| | All | Y/N | Num | Other |
| LMH | 52.01 | 72.58 | 31.11 | 46.96 |
| +CSS | 58.95 | 84.37 | 49.42 | 48.21 |
| +CC | 59.18 | 86.99 | 49.89 | 47.16 |
| +SimpleAug | 53.70 | 74.79 | 34.32 | 47.97 |
| +KDDAug | 59.54 | 86.09 | 54.84 | 46.92 |
| **+ADS-ICT** | **61.03** | **87.94** | **57.02** | **48.29** |

Table 3: Accuracies (%) on VQA-CP v2. Our method can generalize to VQA-Short tasks and consistently improves over base models. ⋆ indicates we only apply ADS-I and I-ICT over the base model.

**UM bias.** We train two separate $\text{UNITER}_B$ models in VCR: a Q-A model taking only questions and answer options as input, and an I-A model taking images and answer options as input. We find that the Q-A model and I-A model can achieve Q2A validation accuracy of $67.20\%$ and $59.28\%$, respectively, which is much higher than random guessing. This validates the existence of shallow mappings inside $(Q, A)$ or $(I, A)$. We extract a subset of data where Q-A and I-A models have more than $90\%$ confidence in predictions and find that $C_c^t$ and $C_c^v$

| Models | VQA v2 val | | | |
|---|---|---|---|---|
| | All | Y/N | Num | Other |
| LMH | 56.34 | 65.05 | 37.63 | 54.68 |
| +CSS | 59.91 | 73.25 | 39.77 | 55.11 |
| +CC | 57.29 | 67.27 | 38.40 | 54.71 |
| +SimpleAug | **62.63** | 79.31 | 41.71 | **55.48** |
| +KDDAug | 62.09 | 79.26 | 40.11 | 54.85 |
| **+ADS-ICT** | 62.40 | **79.55** | **41.93** | 55.29 |

Table 4: Accuracies (%) on VQA v2. Our method can generalize to VQA-Short tasks and consistently improves over base models. ⋆ indicates we only apply ADS-I and I-ICT over the base model.

become extremely high at $78.11\%$ and $64.05\%$.

**DS bias.** Like the identified hypothesis-only bias in text-only benchmarks (Belinkov et al., 2019; Stacey et al., 2020), DS bias enables models to attain the ground-truth labels without visual and question inputs. To verify it, we train an Answer-only model, a $\text{RoBERTa}_B$ (Liu et al., 2019) with only candidate options (both the correct and incorrect answers) as input in VCR, and it achieves $51.84\%$ Q2A accuracy ($69\%$ on SNLI-VE, and $61\%$ on VLEP[5]). This verifies that the DS bias indeed exists. Secondly, using a common feature space[4] (Reimers and Gurevych, 2019), we realize the average intra-sample similarity score between the correct answer and distractors within a sample is 0.31, and the average inter-sample similarity score of every correct answer against its 1000th ranked similar answer candidate (a correct answer from a different question) is 0.34. Moreover, the average intra-sample similarity score among distractors within the same sample is 0.36. This implies that (1) the correct answer can be overly dissimilar to the distractors within the same sample but much more similar to the correct answers to other questions; (2) distractors are also overly similar to each other within the same sample.

## 5.2 Debiased Evaluation

**Debiased Evaluation Setting.** We inference trained I-A, Q-A, and Answer-only models over the VCR validation set and extract samples that meet the following criteria: 1) None of the three models can predict correctly with confidence higher than $25\%$ 2) The correct and incorrect answer choices have a similar number of matched n-grams. We obtain a subset of approximately 2.7K image-text pairs by filtering these conditions, and we consider this subset as a debiased evaluation set, $\underline{\text{VCR}_{\text{Fair}}}$, without direct data augmentation. Lastly, we also

---

[4]https://github.com/UKPLab/sentence-transformers.

apply our ADS method on top of this subset so that, on average, for each original $\{Q, I, A\}$, we can generate four additional types of synthesized data, *i.e.*, $\{I + /-, A + /-\}$. This leads us to obtain around 11K I-Q-A pairs for a domain-shifted evaluation set, $\underline{\text{VCR}_{\text{Adv}}}$. To ensure the integrity of the data, we hired experienced MTurkers to verify the correctness of every synthesized data in $\underline{\text{VCR}_{\text{Adv}}}$[5].

**Bias Mitigation.** We re-analyze UM and DS biases to verify if the dataset biases are mitigated. From Table 5, we observe that correct answers have a much similar frequency of obtaining matching n-grams than distractors against the premise information in $\underline{\text{VCR}_{\text{Far}}}$. This improved balance becomes more noticeable when ADS is applied in $\underline{\text{VCR}_{\text{Adv}}}$. Moreover, the similarity between the correct answers and distractors has also increased, and the distractors become more diverse.

### 5.3 Debiased Training

**Benchmark Comparison.** Results from Table 8 indicate several key observations: (1) ADS-ICT can generalize to various VL models and consistently improve performance across different evaluation settings; (2) The addition of ADS-ICT can bring even much more performance improvement on domain-shifted and debiased settings.

**Other Dataset.** ADS-ICT can generalize to and improve performance over VLU tasks with long candidate options like SNLI-VE, as in Table 8, and even VQA-Short tasks like VQA v2, as in Table 4.

**Debiased Method Comparison.** Despite that former methods lack generalization to VQA-Long tasks, we re-implement former techniques like masking and occlusion boxes, and methods like (Chen et al., 2021; Liang et al., 2020)[5], as in Table 8. To ensure fairness, we even apply ADS-ICT over VQA v2 and VQA-CP v2 for a thorough comparison, as in Table 4. We observe ADS-ICT delivers more significant gains over both datasets.

**Ablation Study.** Table 2 verifies consistent improvement by adding components of our method. Notably, we find that augmenting counterfactual text data can bring greater improvement than factual ones. This also emphasizes the importance of distractors in VQA-Long tasks.

### 5.4 Visual Explainability

We quantitatively and qualitatively verify the models' visual explainability or dependency condi-

---

[5]To save space, more details are presented in the appendix.

|  | $C_c^t$ | $C_d^t$ | $C_c^v$ | $C_d^v$ | $\text{Sim}_{\text{c,d}}$ | $\text{Sim}_{\text{d,d}}$ |
|---|---|---|---|---|---|---|
| VCR$_{\text{Std}}$ | 66.29 | 29.16 | 42.75 | 40.23 | 0.31 | 0.36 |
| VCR$_{\text{Fair}}$ | 48.70 | 32.05 | 42.36 | 40.01 | 0.32 | 0.34 |
| VCR$_{\text{Adv}}$ | 43.93 | 41.09 | 39.28 | 38.94 | 0.35 | 0.33 |

Table 5: Analysis of UM and DS biases on VCR. $\text{Sim}_{\text{c,d}}$ indicates the average semantic similarity between the correct answer and three distractors within a sample and $\text{Sim}_{\text{d,d}}$ indicates the similarity within the distractors only, as in Sec. 5.1.

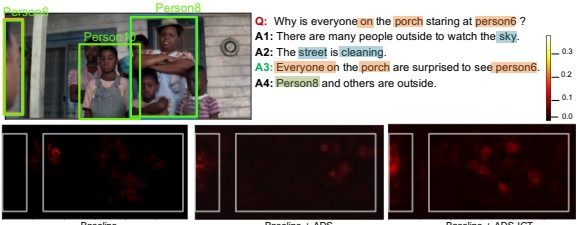

Figure 4: An example from VCR and the paired visual Grad-CAM (Selvaraju et al., 2016) result from a fine-tuned VL-BERT$_\text{L}$ (Su et al., 2019). Based on the image, question, and correct answer, the most relevant entities are "person6" and then everyone on the porch.

| Models | Recall@1 | Recall@2 | Recall@3 |
|---|---|---|---|
| VL-BERT$_\text{L}$ | 46.83 | 59.35 | 67.75 |
| +ADS-ICT | **58.92** | **70.68** | **77.62** |

Table 6: Comparison of recall accuracy(%) for recognizing the most question-related visual objects.

tioning on ADS-ICT. As in Figure 6, the Grad-CAM (Selvaraju et al., 2016) result indicates that the base model ignores the most relevant entity, "person6". However, after adding ADS-ICT, the model relies more on the relevant regions like "person6" and "person8". We further calculate the recall accuracy of the model for retrieving the most relevant entities by comparing its attention values against object labels[5]. As in Table 12, we observe that the recall accuracy is significantly increased with ADS-ICT[5], indicating that the model's visual explainability (dependency) has improved.

## 6 Time Consumption

Our coarse-to-fine region removal method is flexible and generalizable as the number of runs of the image inpainting process can be adjusted depending on the scenarios to decrease the time consumption. After selecting the region to be removed, our proposed coarse-to-fine region removal will be conducted by two main steps: 1) Initial one-pass full region removal/inpainting by $SPL_p$; 2) Triple-pass autoregression region removal/inpainting by $SPL_f$. The triple-pass autoregression strategy en-

| Exp Index | | # Runs of | | | | Time Consumption ($ms$) | Accuracy (%) |
|---|---|---|---|---|---|---|---|
| | $SPL_p$ | $SPL_f$ $(O+M+N)$ | $1st$ **Pass** $(O)$ | $2nd$ **Pass** $(M)$ | $3rd$ **Pass** $(N)$ | | |
| 1 | 0 | 0 | 0 | 0 | 0 | $10^1$ | 76.07 |
| 2 | 1 | 0 | 0 | 0 | 0 | $10^2$ | 76.18 |
| 3 | 1 | 1 | 1 | 0 | 0 | $2 \times 10^2$ | 76.30 |
| 4 | 1 | 14 | 1 | 3 | 9 | $1.3 \times 10^3$ | 76.51 |
| 5 | 1 | 21 | 1 | 4 | 16 | $2.2 \times 10^3$ | 76.88 |
| 6 | 1 | 30 | 1 | 4 | 25 | $3.5 \times 10^3$ | 76.87 |
| 7 | 1 | 26 | 1 | 9 | 16 | $3 \times 10^3$ | 76.88 |

Table 7: Time consumption comparison among variations of our coarse-to-fine region removal method. During this comparison study, the base model is a VL-BERT$_L$ (Su et al., 2019) running on one NVIDIA TITAN RTX GPU of 24 GB. The input image is of size $224 \times 224$.

sures fine-grained images and avoids superficial artifacts, as addressed in prior works. After the $1st$ run of full region inpainting by $SPL_f$, we split the region into smaller $M$ regions evenly and run the inpainting process for each smaller region, respectively. A similar procedure applies to the third pass of $N$ runs. Triple-pass autoregression is a flexible solution, as both M and N can be set to any arbitrary positive integers ($2 \leqslant M \leqslant N$) depending on the situation to decrease the overhead time consumption. Hence, The total runs of the inpainting process in the tr iple-pass strategy is $1 + M + N$.

Based on Table 7, we have four observations: 1) As in Exp 1, If we do not conduct coarse-to-fine region removal (neither of the two steps will be applied), we essentially apply an occlusion box over the selected region. This is the same approach as the prior works and will cause a relatively low downstream QA accuracy of 76.07 %; 2) Comparing Exp 1-3, we observe that adding the image inpainting process by $SPL_p$ and $SPL_f$ will both consistently improve the downstream performance; 3) Based on Exp 3-5, we can see a performance improvement trend as we apply more refined region inpainting via increasing the value of $M$ and $N$; 4) Our method is tolerant to the variations of $O$, $M$ and $N$ and can still achieve noticeable performance improvement even when the overall number of runs is low as 1 with time consumption of $2 \times 10^2 ms$ for one image. In practice, we found that setting $M$ and $N$ to 4 and 16, respectively, generally achieves optimal performance while maintaining reasonable inference time consumption of $2.2 \times 10^3$ ms for one sample image However, as mentioned, the triple-pass autoregression strategy is a flexible and general solution. Thus, $O$, $M$ and $N$ can be adjusted according to the actual situation to decrease time consumption and our method can still provide similar results.

## 7 Conclusion

This paper analyzed dataset biases and their underlying causes in VQA-Long. Our findings shed light on how these biases can impact the evaluation of VL models and the importance of mitigation. We hope our work will inspire future research in developing rigorous data annotation processes and strategies to mitigate the influence of dataset biases.

## Limitations

First of all, our proposed method, ADS-I, is designed to remove pertinent parts of visual regions to generate synthetic factual images, I+, and irrelevant regions to create I-. We adopted techniques from an existing study (Chen et al., 2020) to accomplish this. However, some noise still persists, which might impact the accuracy of determining the relevant region. A promising next step might involve enhancing the quality of the generated images by addressing these noise issues.

Besides, to ensure the high quality of our constructed debiased evaluation benchmarks, we opted for manual verification, which consequently increased the overall cost of our research study. We anticipate that some cost-efficient yet reliable pre-selection procedure could be developed to mitigate these costs. Additionally, the manual selection process could introduce a certain level of subjectivity into the dataset, which needs to be considered.

## Ethics Statement

ChatGPT is pre-trained on the colossal corpus which is likely to contain potential racial and gender bias. Therefore, if someone finds our work interesting and would like to use it in a specific environment, we strongly suggest the user check the potential bias before usage. In addition, it is hard to control the generation of LLMs like ChatGPT. We

should be aware of the potential problems caused by incorrect predictions.

## Acknowledgements

This work is supported by DARPA MCS program under Cooperative Agreement N66001-19-2-4032.

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

# A Appendix

## A.1 Difference between VQA-Short (Classification) and VQA-Long (MCQ)

Visual Question Answering (VQA) (Antol et al., 2015; Zellers et al., 2019; Lei et al., 2020) is a popular VLU task where the premise information[6] is provided, and the objective is to answer the question correctly. It is challenging as it requires reasoning with the integrated information from visual

---

[6]In this paper, "premise" refers to the given context or arguments. For text-only QA, the premise is the question. But for VQA, it consists of both the image and the question.

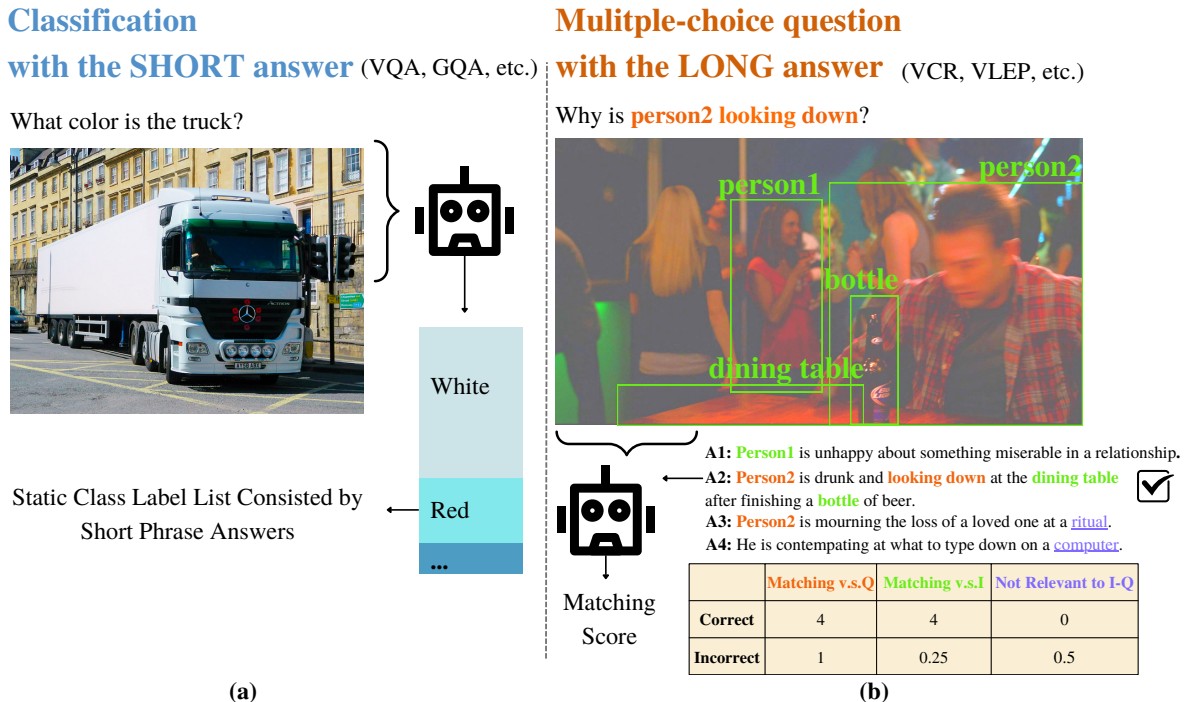

Figure 5: (a): Prior studies examined dataset bias in the distribution of short-phrase answers (*e.g.* "white" is often the answer when asking about color). (b): Our work investigates the biases in VQA with long answer choices, where the correct answer has more n-grams overlapped with the image and question (as shown in BurntOrange and ForestGreen). Meanwhile, the incorrect answers contain more irrelevant n-grams to the scene (as shown in purple).

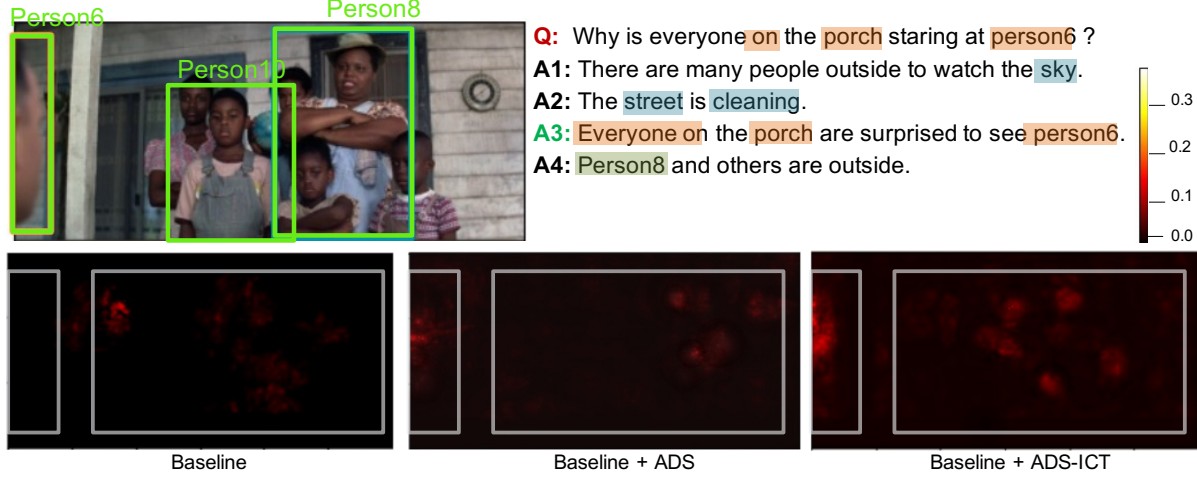

Figure 6: An example from VCR and the paired visual Grad-CAM (Selvaraju et al., 2016) result from a finetuned VL-BERT$_L$ (Su et al., 2019). Based on the image, question, and correct answer, the most relevant entities are "person6" and then everyone on the porch. With training, we expect the VL model to integrate multimodal information and commonsense when selecting the correct answer (labeled by a green check). For instance, to answer this question, we expect the model to focus on [person6] on the left and people in the center. However, the model fails to pick up the correct visual clue and focuses on the irrelevant entity, the window in the background. The orange words are the overlapping words between the correct choice and the question.

and text modalities. Most of the existing VQA benchmarks conventionally use either of the two settings: 1) **Classification**: As in the top of Figure 5, in this setting, different questions share one set of context-free answer choices, *e.g.* VQA v2 (Goyal et al., 2017), GQA (Hudson and Manning, 2019), and OK-VQA (Marino et al., 2019), and models are supervised for classification using the shared answer choices as fixed class labels. Accordingly, to generate a common answer dictionary, the answer choices are limited to be "short" words or phrases, and the questions tend to focus on visual perception. 2) **Multiple-Choice Question (MCQ)**: As in the bottom of Figure 5, every question has a unique set of context-dependent answer choices, *e.g.* four sample-specific sentence-level answer choices in VCR. Models are trained with a generalized objective, cross-modality matching, predicting the degree of match between each answer choice and the premise information. This is similar to other VLU tasks, such as visual reasoning (Suhr et al., 2018) and visual entailment (Xie et al., 2019). Due to this flexibility, the questions tend to be diverse, highly semantic, and centering around human activities , *e.g.* VCR (Zellers et al., 2019), VLEP (Lei et al., 2020), MovieQA (Tapaswi et al., 2016), TVQA (Lei et al., 2018), and TVQA+ (Lei et al., 2019). Accordingly, the answer choices tend to be "long" and include rich context, and the images are more complex with multiple entities and diverse scenarios. The second setting is analogous to the MCQ problems humans face and, thereupon, can more likely generalize to real-world scenarios. Recently, the field has advanced towards generating highly semantic and challenging VL benchmarks mimicking real-world challenges, and the second setting is becoming increasingly important.

Although recent VL models (Li et al., 2023, 2022; Wang et al., 2022a) can achieve good performance on many above-mentioned benchmarks, they still suffer from bias issues. Many studies have discussed the bias issues in Classification-VQA tasks (Chen et al., 2020a; Dancette et al., 2021; Gokhale et al., 2020; Gupta et al., 2022; Niu et al., 2021; Ramakrishnan et al., 2018), but none of them investigate in the MCQ setting, with the exception of a recent study (Ye and Kovashka, 2021). In the classification setting, as in VQA v2, supervised models do not actually see the answer choices at all in the input but only the images and questions. In other words, they are trained to classify image-

question pairs into a fixed list of class labels consisting of short-word answers. Hence, they can easily divert to capture shallow correlations focusing on different question types and their mapped label distributions for it is easier for models to learn shallow correlations within the same modality, text. As the analysis at the top of Figure 5, we find that questions starting with "What color is the" will often be classified to the label, "white" regardless of the actual context in VQA v2 (Agrawal et al., 2018).

However, these bias problems may not apply to the MCQ setting. Some of the main reasons include the following: The questions are much more diverse and difficult to be categorized into basic types; The answer choices are not only dynamic and context-dependent but also contain much more information even than the questions. Thereupon, they have to be in the input not as fixed class labels. Yet, in this work, we argue that, in the MCQ setting, some other bias-related problems are more likely to form and supervised models are more vulnerable . With more input types, *i.e.* (Image, Question, Answer), more implicit correlations in between may likely form. On the other hand, the MCQ setting requires generating long context-dependent incorrect choices (distractors) for each sample, and this further brings in much more complexity and artifacts. As verified by our study, we discover that bias-related problems in the MCQ setting are centered around the answer choices and their relationships against the premises. In addition, we believe it is very important to encourage future works to focus on analyzing the bias problems within VLU tasks with the MCQ setting.

The former work (Ye and Kovashka, 2021) points out that the correct answers have more exact matching entities (mainly pronouns like "he/she, 'they', *etc.*) against the questions in VCR, referred to the bottom of Figure 5 where the orange-highlighted text represents the overlapping entities against the question. Despite how shallow this may seem, we verified that a simple heuristic rule of picking the choice with the most overlapping entities could deliver 66.15% Q2A accuracy and 63.04% QA2R accuracy on VCR. Nevertheless, (Ye and Kovashka, 2021) is mainly constrained to resolve the explicit advantage of the correct answers containing more exact matching pronouns in the text modality solely (between question and answer choices) of VCR. Therefore it does

not investigate and generalize to problems about the correlation among general components in the text, other cross-modal correlations between the premise and ground-truth annotations(, *e.g.* the green-underscored text at the bottom of Figure 5), other VLU benchmarks with the MCQ setting and, most importantly, the fundamental problems centering around the generation process of distractors.

To fix these issues, we first conduct a comprehensive analysis across several VLU benchmarks *i.e.* VCR (Zellers et al., 2019), SNLI-VE (Xie et al., 2019), and VLEP (Lei et al., 2020). Among them, we identify two common but severe problems: (1) The Unbalanced Matching (UM) problem: Answer choices have an unbalanced matching of n-grams against the premises. This includes the problem of the correct answers containing more matching nouns, adjectives, *etc.* and more n-grams like phrases. On the other hand, distractors have not only fewer matching n-grams but also much more n-grams not related to the scene at all; (2) The Answer-only Bias problem: We realize that distractors are often generated without sufficient visual premise as the correct answers, filtered by simple metrics and modified by the same set of heuristic rules, which leads to lack of diversity and frequent usage of certain words or phrases. Consequently, distractors are excessively dissimilar to the correct answers and over-similar to other distractors within the same sample. We systematically analyze these problems and their causes backed with statistical evidence and further conduct experiments to demonstrate their effect on models' learning of biases. Furthermore, due to the bias-related problems heavily centering around answer choices and distractors also tend to reference entities or concepts not related to the scene, we discover that existing VL models may under-utilize visual information and struggle to learn query-related visual dependency, as shown in Figure. 6.

The former data synthesis approaches (Chen et al., 2020a; Dancette et al., 2021; Gokhale et al., 2020; Gupta et al., 2022; Niu et al., 2021; Ramakrishnan et al., 2018) fail to address the bias problems in the MCQ setting. Moreover, they either only focus on the text modality, cannot resolve problems related to long-sentence distractors, or severely disturbed the data distribution via superficial masking on text or occlusions on images, as in the right of Figure 2. Differently, to assist models in countering the bias problems centering around answer choices and under-utilization of visual information leading to incorrect visual dependencies, we propose a novel Adversarial VL Data Synthesis (ADS) method consisting of ADS-T to generate synthetic factual and counterfactual text data and ADS-I for image data. ADS-T can directly assist the generation of long-sentence answers and distractors, and ADS-I can generate synthesized images closer to real images with minimized data distribution via the semantic focus of the query. Additionally, former debiasing methods (Chen et al., 2020a; Dancette et al., 2021; Gokhale et al., 2020; Gupta et al., 2022; Niu et al., 2021; Ramakrishnan et al., 2018) directly augment the synthesized counterfactual questions and images, as $I_-$ and $Q_-$ in Figure. 2, in the input for model training, as they do not need to find explicit paired answers. However, this does not apply to the MCQ setting because explicit answer choices are required as the paired input data for $I_-$ and $Q_-$. Nonetheless, it is impossible to find paired answer choices without additional manual annotations, as referred to $I_-$ and $Q_-$ in Figure. 2. In this work, we successfully resolve this challenging problem by proposing ICT strategy to utilize the synthesized counterfactual data in the MCQ setting. Lastly, with human verification, we employ ADS to create domain-shifted evaluation benchmarks based on VCR to test existing and future VL models' robustness. Although we primarily present analysis and experiments across VCR, SNLI-VE, and VLEP benchmarks, our identified problems and proposed solutions can be generalized to other VLU tasks with rich context.

### A.2 Implementation Details

For object labels in VCR, we combine the provided annotated object labels with the generated ones as in (Anderson et al., 2018). When training $SPL_f$ on downstream dataset, *e.g.* VCR(Zellers et al., 2019), we skip visual regions whose area is smaller than 1/64 of the whole image. In Eq. 1, $\lambda$ is set to be 1 and $\alpha$ would be set to be 0.4.

In training with ICT over our synthesized data, for each $(I, Q, A)$ pair, with ADS, each image would result in an average of one positive and one negative synthesized image; The original four answer choices would also result in one synthesized positive answer choice and three synthesized negative answer choices. As shown in Fig. 7, every original $(I, Q, A)$ sample data would result in seven additional samples for training with classification

loss and three for contrastive loss.

## A.3 Multimodal Distractor Refinement.

After obtaining more diverse candidate answers from Multimodal Distractor Generation, we conducted an in-house study of over 100 samples to test how humans can polish those candidate answers into quality distractors. We determine the criteria of quality distractors in each sample of VQA-Long to be: (1) Similar to the correct answer and containing a comparable amount of matched n-grams; (2) Relevant to the given context and not containing irrelevant n-grams to the scene; (3) Not false negative to cause ambiguity; (4) Having descent diversity to be different from each other. With those criteria, for each sample, we have a group of three experienced annotators and ask each to annotate three quality distractors. Following, for each annotated sample, we ask another group of 2 experienced annotators to rate the distractors based on the four criteria and send back the unqualified ones to the first group to redo the annotation until they suffice. For each sample, we provide annotators rich references as the input information, including (1) The original sample image, question, and answer; (2) The retrieved available caption for each image; (3) The matched n-grams between the correct answer and the premise information; (4) The top 10 most salient extracted object labels; (5) The top 10 ranked answer candidates by Multimodal Distractor Generation.

With this study, we realize that quality distractor generation is challenging and much harder than answer generation, as it requires complex commonsense reasoning and knowledge. Thus it is difficult to rely on one or two noisy scores or heuristic rules as the former solutions to secure quality distractors. With these insights, we employ a largely pre-trained ChatGpt (OpenAI, 2023) and expect it to mimic the generation process of humans. Accordingly, we provide the input with rich references to the model as to humans via the prompt. We also handpicked 30 human-annotated examples and each time randomly select 5 of them as the examples. Thence ChatGPT has the five types of references as the input and the three qualified distractors as the output. With sufficient multimodal references and examples as constraints, we find this approach effective in generating quality distractors with the two biases mitigated and can apply for large-scale annotations.

## A.4 Coarse-to-Fine Region Removal

After determining the relevant regions in images, following the structure of SPL (Zhang et al., 2021a), we design a framework to remove relevant and irrelevant regions to generate realistic natural images, **I+/-**. In this framework, we reserve two $SPLs$, an $SPL_p$ pretrained on Places2 (Zhou et al., 2017) as in (Zhang et al., 2021a) and another $SPL_f$ finetuned on VCR. We first retrieve the segmented polygons (if the entity is provided by VCR annotation) or bounding boxes (if generated) of the selected relevant entities. After determining the dimensions of the maximum inscribed rectangles within the polygons or boxes, we create corresponding rectangle maskings in ratios, $(0.7, 0.5, 0.3)$ of the maximum dimensions. Similarly, we also calculate the maximum dimensions of inscribed rectangles in regions that have no entity overlapped on top at all and create maskings of different ratios within those regions. When fine-tuning $SPL_f$ on the VCR training set, we input images with regions masked by one of those maskings and supervise $SPL_f$ to reconstruct the masked region. Therefore, essentially $SPL_f$ is trained to reconstruct the interior of either an entity or an open background region based on its neighboring non-masked pixels and patterns. In order to create **I+**, in inferencing, we filter to irrelevant regions, $R(s_i) \notin$ R E L and create maskings of the minimum circumscribed rectangles around the boxes or polygons. We then feed images masked by those maskings into SPLs to remove the irrelevant entities. Similarly, For creating **I-**, we create similar maskings over the relevant regions from $REL$ to inference to remove the relevant entities. Some examples of the reconstructed images by $SPL$ are shown in 7.

In order to produce fine-grained images and avoid drastically disturbing existing image distribution and bringing obvious artifacts/biases as occlusion boxes do in (Chen et al., 2020a; Liang et al., 2020; Gokhale et al., 2020), in practice, during inferencing, we apply a coarse-to-refine strategy by first passing the masked images into $SPL_p$ that was pretrained on a larger dataset and then feed the reconstructed output to $SPL_f$ that was fine-tuned more specifically to refine the reconstruction. As inferencing in $SPL_f$, we additionally refine the image via a triple-grid mechanism. As in Fig. 3, for a given masked region, we evenly split it into $M$ blocks and $N$ blocks respectively where $2 < M < N$. In the first pass, we allow $SPL_f$ to

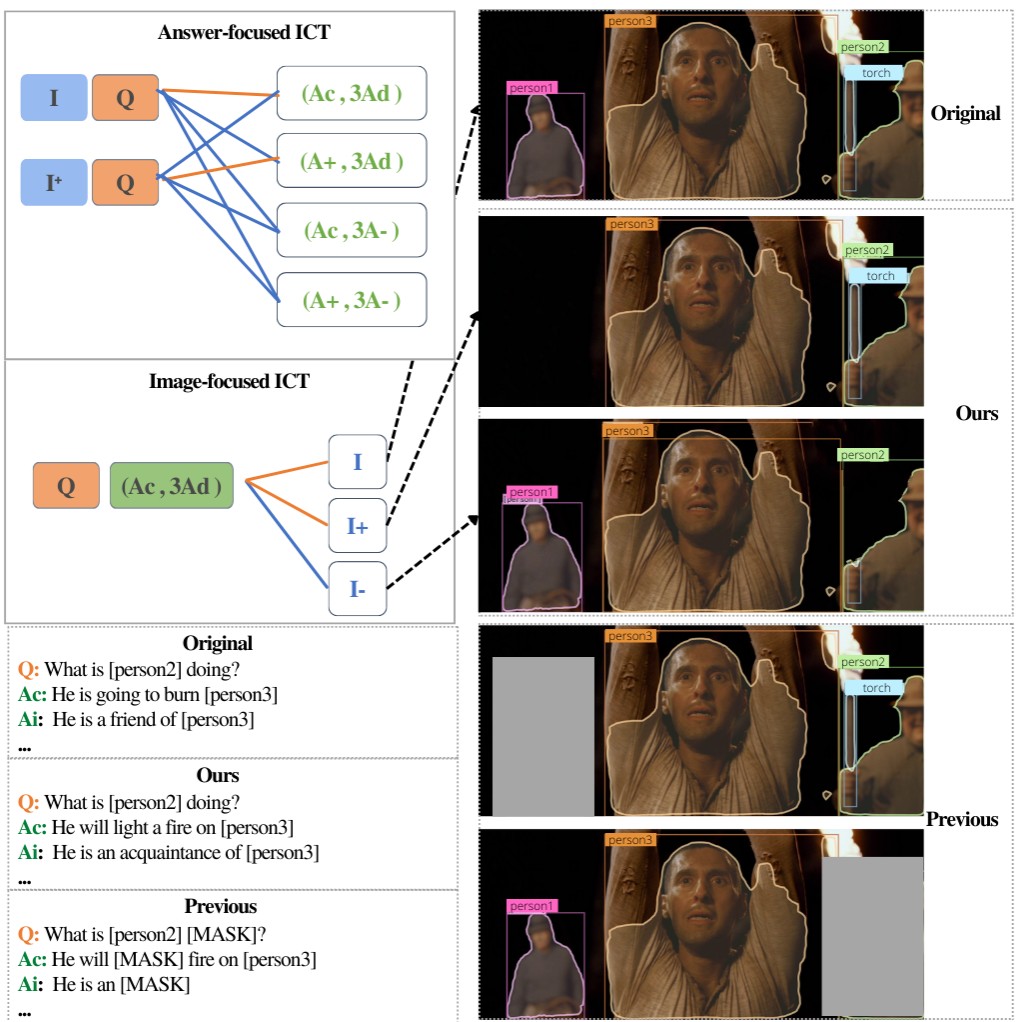

Figure 7: Diagram of all the combinations of (I, Q, A) pairs utilized in training. The pairs from the top block are utilized in QA classification training. The pairs in the bottom block are used in intra-sample contrastive learning. The orange lines indicate the overlapping/repeated pairs. In practice, we only pass those pairs once in the feedforward operations. $Ac$ represents the correct choices. $Ai$ represents the incorrect choices.

reconstruct the whole masked region in one pass and then revisit the same region with smaller maskings in the following two passes. In the second pass, we take turns to turn each of the $M$ blocks in order into a smaller masking, from the top left to the bottom right, and accordingly reconstruct each masked region to refine. Note that when we reconstruct the first block from the top left of $M$ blocks, the visual regions of the rest $M-1$ blocks are not masked but in-painted with results from the former pass as placeholders. Therefore, we cumulatively inference $M$ times to refine the whole region in the 2nd overall pass. A similar procedure is carried out for the third pass with $N$ blocks. This method allows the framework to maintain the global consistency in reconstructed visual patterns while obtaining the flexibility in refining smaller regions.

## A.5 Domain-shift and Debiased Evaluation Benchmarks

We first adopt the formerly published evaluation setting via changes of pronouns (Ye and Kovashka, 2021), which we refer to as $\underline{\text{VCR}_{\text{Pronoun-Shift}}}$. Further, we then use previously finetuned stylish models, I-A, Q-A, and A-only models, to perform inference on the standard VCR validation set. Using the confidence scores of these models, we extract samples that meet the following criteria: 1) Neither the I-A nor Q-A model can predict correctly with confidence higher than $50\%$ over these samples, 2) the A-only model also cannot predict correctly with confidence higher than $25\%$, and 3) the correct and incorrect answer choices have a similar number of matched n-grams. Filtering using these three conditions, we get a subset of approximately 2.7K image-text pairs. This adversarial filtering mitigates the effect of UM and Answer-Only biases on this subset, and we consider it as a debiased evaluation setting, referred to as $\underline{\text{VCR}_{\text{Fair}}}$ without direct domain shift.

Lastly, we apply our ADS method on top of this subset so that, on average, for each original $\{Q, I, (A_c, A_d)\}$, we can end up with four combinations of synthetic data, *i.e.* $\{Q, I, (A_+, A_-)\}$, $\{Q, I_+, (A_c, A_d)\}$, $\{Q, I, (A_+, A_d)\}$, $\{Q, I, (A_c, A_-)\}$. This leads us to obtain around 11K I-Q-A pairs for another domain-shift evaluation, $\underline{\text{VCR}_{\text{Adv}}}$. To ensure the integrity of the data, we hired experienced Amazon Turkers to verify the correctness of every synthesized data in $\underline{\text{VCR}_{\text{Adv}}}$ to avoid any superficial errors and artifacts.

## A.6 Benchmark Evaluation

Referring to Table. 8.

## A.7 Problem with Lexical Similarity in Adversarial Matching

The difficulty of generating long sentence distractors is prominently higher than short noun distractors. For facilitating it, Adversarial Matching (Zellers et al., 2019; Williams et al., 2022; Lei et al., 2020), where positive answers are recycled to serve as negatives for other questions, has been recently adopted as a popular method. It generally leverages the difference between two language models' predicted probabilities to represent the relevance score, $S_{\text{rel}}(p, r)$ of a response $r$ against the text premise $p$ and the similarity score, $S_{\text{sim}}(c, r)$ between the response $r$ against the correct answer $c$. The difference between these two would be used for the final ranking, $\lambda\left(f(S_{rel}(p, r)) - \alpha \cdot g(S_{sim}(c, r))\right)$, to ensure the selected distractors dissimilar to the correct answer but relevant to the text premise, where $\lambda$ and $\alpha$ are hyper-parameters and $f$ as well as $g$ are fixed functions.

Despite the claimed effectiveness, profound problems exist with this method. **Insufficiency of Scores:** First of all, the two scores come from different sources and are not normalized, thus obviously causing integrity. Secondly, the scores are not fine-grained and semantically reliable, as in Fig. 8, for the false positives, despite the fact they may possess critical conflicts against the text premise, those high-quality distractors would be eliminated. In contrast, the first false negative, paraphrasing the original text premise, incorporates the exact meanings but is deemed less "similar." To avoid false negatives, in practice, the second term, $\alpha \cdot g(S_{sim}(c, r))$ would be profoundly emphasized, resulting in a specific selection window favoring lower-ranked candidates and unavoidably eliminating high-quality distractors. Owing to the same window's origin, these distractors lean to share similar traits, especially after the similar heuristic modification with templates.

## A.8 Bias Analysis

As shown in Table. 9, it is obvious that the identified two types of biases, UM and Answer-only biases are common across benchmarks. With the aid of ADS, we observe significant improvement

| Model | VCR$_{\text{Std}}$ | | | VCR$_{P-\text{shift}}$ | VCR$_{\text{Fair}}$ | VCR$_{\text{Adv}}$ | SNLI-VE | |
| | Q2A | QA2R | Q2AR | Q2A | Q2A | Q2A | Val | Test |
|---|---|---|---|---|---|---|---|---|
| Heuristics-Only | 66.29 | 65.98 | 43.74 | 49.75 | 48.70 | 43.93 | 69.77 | 69.30 |
| VL-BERT$_L$ | 75.51 | 77.95 | 58.86 | 71.13 | 72.84 | 70.46 | 74.66 | 74.02 |
| VL-BERT$_L$ + ADS-ICT | **77.33 [+1.82]** | **79.93 [+1.98]** | **61.80 [+2.94]** | **74.26 [+3.13]** | **76.12 [+3.28]** | **73.72 [+3.26]** | **76.27 [+1.66]** | **76.33 [+2.31]** |
| UNITER$_L$ | 76.72 | 80.01 | 61.38 | 73.84 | 74.99 | 72.48 | 79.02 | 79.19 |
| UNITER$_L$ + ADS-ICT | **78.23 [+1.51]** | **82.29 [+2.27]** | **64.37 [+2.99]** | **76.81 [+2.97]** | **77.36 [+2.37]** | **74.74 [+2.26]** | **80.14 [+1.12]** | **80.23 [+1.04]** |
| VILLA$_L$ | 78.28 | 82.20 | 64.34 | 75.43 | 77.01 | 74.05 | 79.64 | 79.32 |
| VILLA$_L$ + ADS-ICT | **78.89 [+0.61]** | **82.77 [+0.57]** | **65.30 [+0.96]** | **76.80 [+1.37]** | **77.75 [+0.74]** | **75.38 [+1.33]** | **80.87 [+1.23]** | **80.28 [+0.96]** |

Table 8: Our comparisons against benchmark methods are based on our own re-implementation. QA accuracy is adopted for VCR-related benchmarks over Q2A, QA2R, and Q2AR tasks. For the SNLI-VE benchmark, we use accuracy based on classification over three labels: entailment, neutral, and contradiction. The results of Heuristics-Only are obtained by taking the best performance from a mix of heuristic rules utilizing the two biases. For example, the method always selects the option with the most matching n-grams

| Dataset | | w/o ADS | | | w/ ADS | | |
| | | Percentage of Highest Matching Tokens | | Avg. Matching Tokens | Percentage of Highest Matching Tokens | | Avg. Matching Tokens |
| | | v.s. Question | v.s. Image | v.s. Question-Image | v.s. Question | v.s. Image | v.s. Question-Image |
|---|---|---|---|---|---|---|---|
| **VCR** | Correct | 66.29 | 42.75 | 2.02 | 43.93 | 39.28 | 1.95 |
| | Inoccrect | 29.16 | 40.23 | 1.8 | 41.09 | 38.94 | 1.9 |
| | | v.s. Caption | v.s. Image | v.s. Caption-Image | v.s. Caption | v.s. Image | v.s. Caption-Image |
| **SNLI-VE** | Entail | 69.77 | 57.77 | 4.11 | 57.41 | 52.59 | 3.82 |
| | Contradict-Neutral | 45.4 | 36.16 | 3.23 | 48.93 | 39.72 | 3.63 |
| | | v.s. Subtitles | | | | | |
| **VLEP** | Correct | 48.85 | | 12.5 | 43.17 | | 11.43 |
| | Inoccrect | 36.19 | | 11.76 | 39.01 | | 11.26 |

Table 9: Bias Analysis over three benchmarks with ADS

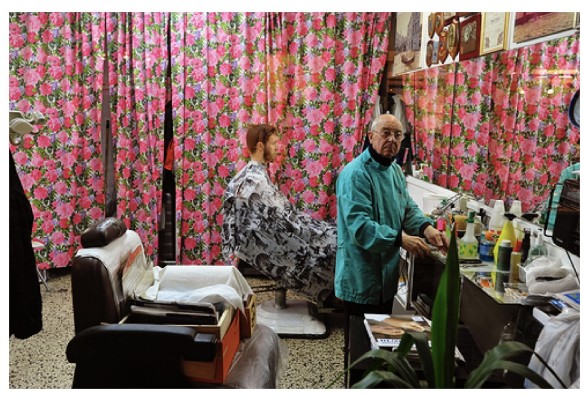

| | | Similarity |
|---|---|---|
| **Original Text** | A Barber with glasses is standing at his counter while another man is waiting in the barber chair. | 1 |
| **False Positives** | A Barber with glasses is standing at his counter while another **woman** is waiting in the barber chair. | 0.921 |
| | A Barber with **sunglasses** is standing at his counter while another man is waiting in the barber chair . | 0.914 |
| | A Barber with **computer glasses** is standing at his counter while another man is waiting in the **gaming** chair . | 0.903 |
| **False Negatives** | A **male customer** is sitting at a chair waiting for the **haircut stylist** with glasses standing **next to him** | 0.635 |
| | An **indoor scene of two** men, **one of which looks younger with mustache** while the **other seems older** with a **bald head**. | 0.361 |

Figure 8: Analysis over the lexical similarity between hypotheses (answer choices) against an example image from SNLI-VE.

| Fashion Model | Dataset | w/o ADS | | w/ ADS | |
|---|---|---|---|---|---|
| **A-Only** | VCR | 51.84 (Q2A) | 58.39 (QA2R) | 46.52 (Q2A) | 52.53 (QA2R) |
| | SNLI-VE | 69.36 | | 54.22 | |
| | VLEP | 61.18 | | 53.37 | |
| **QA-Only** | VCR | 67.21 | 69.4 | 61.98 | 63.89 |
| | SNLI-VE | 76.93 | | 68.66 | |
| **IA-Only** | VCR | 59.28 | 64.02 | 56.38 | 62.95 |

Table 10: Evaluation of fashion models over benchmarks with ADS. A-Only stands for answer-only. QA-Only stands for Question-Answer-Only. IA-Only stands for Image-Answer-Only.

| Dataset | | $w/o$ ADS | $w/$ ADS |
| | | # Irrelevant n-grams v.s. Premise | |
|---|---|---|---|
| **VCR** | Correct | 0.18 | 0.18 |
| | Incorrect | 1.14 | 0.8 |
| **SNLI-VE** | Entail | 0.15 | 0.15 |
| | Contradict - neutral | 1.49 | 0.74 |
| **VLEP** | Correct | 0.68 | 0.68 |
| | Incorrect | 1.03 | 0.75 |

Table 11: Analysis of irrelevant n-grams over benchmarks with ADS

over the overlapping n-grams between the correct answer choice and the incorrect ones against the premise information (including both visual and text premise information).

If we further re-train the three fashion models across the three benchmarks as in Table. 10, the performance of the three fashion models all drop consistently. This further verifies the mitigation effect of ADS over the two biases.

As in Table 11, we conduct a preliminary experiment by first hiring three experienced annotators and randomly sampling 100 samples for each of the three benchmarks. For each sample with the visual and text premise (image and question in VCR) and the answer choices, we extract all the possible n-grams from the answer choices. Following this, we ask each annotator to judge how many of the n-grams are relevant to the visual and text premise information. The overall averaged result is presented in Table 11.

**Distractors Have More Irrelevant n-grams.** In addition to lower n-gram overlap with the premise, we observed that distractors also contain more n-grams that are irrelevant to the given context. As shown in Figure 11, some n-grams, like "ritual" and "computer", have no clear association with the image or question. Our analysis, employing both human and pre-trained models, reveals that, on average, distractors possess 1.14 n-grams that are irrelevant to the premise, which is significantly higher than the 0.18 n-grams for the correct answer

**Distractors with Limited Diversity.** Since distractors tend to be generated without visual premise information in AM (Lei et al., 2020; Zellers et al., 2019; Li et al., 2020) but only the text, there is limited information as the reference for selecting diverse distractors. This becomes more severe in cases where the text premise has scarce information *e.g.*, the questions with short stems in VCR like "Why is person2 looking down?" in Figure 5 or "What is going to happen next?". Besides, in AM, the top selected candidate answers would then be modified based on heuristic rules, evolving token-level replacement, and these invariant rules can further lead to over-similar production. On the other hand, without being given the image in the manual annotation in (Xie et al., 2019; Do et al., 2020; Kayser et al., 2021), annotators are forced to come up with imaginary content to add to the distractors. The diversity solely relies on the annotators' discipline and imagination abilities with no constraints, and this, unfortunately, leads to lazy annotations with distractors containing repeated and irrelevant n-grams.

## A.9 Visual Explainability

Figure 6 visualizes the Grad-CAM (Selvaraju et al., 2016) results of a base model. The Grad-CAM result of the baseline model indicates that it does not pay attention to the most relevant entity, "per-

son6". After adding ADS, the model appears to rely more on the regions of "person6" and "person8". If we further apply ICT, the model demonstrates a more obvious (confident) visual dependency over the most relevant entities. To quantify the improvement in the model's visual explainability and its ability to capture correct visual dependency, we calculate the recall accuracy of base models for recognizing the most question-related visual objects. We retrieve the grounded entities in the questions provided in VCR as the ground-truth labels, extract the attention values of base models over each visual token in the last hidden layer's output, and compare the attention values against the labels to calculate the recall accuracy. As shown in Table 12, we observe that the recall accuracy is significantly increased with ADS-ICT[5], indicating that the model's visual explainability has improved, and it has learned the correct visual dependency.

| Model | Recall@1 | Recall@2 | Recall@3 |
|---|---|---|---|
| $VL\text{-}BERT_L$ | 46.83 | 59.35 | 67.75 |
| $VL\text{-}BERT_L$ + ADS-ICT | **58.92** | **70.68** | **77.62** |
| $UNITER_L$ | 49.93 | 66.86 | 71.55 |
| $UNITER_L$ + ADS-ICT | **60.90** | **75.93** | **80.47** |

Table 12: Visual explainability. We calculate the recall of models by retrieving the most relevant entities in response to a given image, question, and answer pair.