# OpenReview forum: "Dataset Bias Mitigation in Multiple-Choice Visual Question Answering and Beyond"
_EMNLP/2023/Conference — EMNLP 2023 Findings_

### Official Review · Reviewer_32Fg · 2023-08-01

**Soundness:** 4

**Excitement:**

3: Ambivalent: It has merits (e.g., it reports state-of-the-art results, the idea is nice), but there are key weaknesses (e.g., it describes incremental work), and it can significantly benefit from another round of revision. However, I won't object to accepting it if my co-reviewers champion it.

**Paper Topic And Main Contributions:**

This paper focuses on the analysis and mitigation of dataset biases in Visual Question Answering (VQA) tasks with long answer choices. The authors identify two types of biases, Unbalanced Matching bias and Distractor Similarity bias, and propose an Adversarial Data Synthesis (ADS) method and an Intra-sample Counterfactual Training (ICT) method to mitigate these biases. The proposed methods are evaluated on multiple benchmarks and demonstrate consistent improvements in model performance.

**Questions For The Authors:**

1. Can you provide the time consumption of your proposed method with other baselines?

**Reasons To Accept:**

1. The paper addresses an important issue, i.e., dataset bias in VQA tasks, and provides a different perspective to mitigate these dataset biases.

2. The proposed ADS method effectively synthesizes factual and counterfactual data to mitigate biases. Meanwhile, the ICT method enhances models' training by focusing on intra-sample differentiation.

3. The paper is well-structured and provides extensive empirical results to demonstrate the effectiveness of ADS and ICT in consistently improving model performance across different benchmarks.

**Reasons To Reject:**

1. The synthetic factual image generation will inevitably introduce noises, and controlling the quality of the synthetic images need extra human supervision.

2. The overall inference process of this method seems to demand a higher time consumption. There is a lack of in-depth analysis regarding the comparison of time consumption with other baselines.

**Reproducibility:**

4: Could mostly reproduce the results, but there may be some variation because of sample variance or minor variations in their interpretation of the protocol or method.

**Reviewer Confidence:**

4: Quite sure. I tried to check the important points carefully. It's unlikely, though conceivable, that I missed something that should affect my ratings.

---

> ### Author Rebuttal · Authors · 2023-08-29
>
> **Noises in Synthetic Images:**
> Compared with prior works, our method actually tremendously decreases the artificial noises and minimizes the disturbance of image distribution. Specifically, unlike previous solutions of naively applying occlusion boxes, we innovatively create factual and counterfactual images via image inpainting. Even trivial noises may still exist, but the apparent performance improvements by augmenting our synthetic images in training over multiple base models can already prove the quality and value of our generated images. Meanwhile, we also conducted human verification over the generated synthetic images (3 annotators for each sample) in the validation set to ensure the evaluation quality further.
>
> **Time Consumption:**
> Our coarse-to-fine region removal method is flexible and generalizable as the number of runs of image inpainting process can be adjusted depending on the scenarios to decrease the time consumption. In practice, we found that setting M and N to 4 and 16, respectively, generally achieves the optimal performance. The time consumption is 2.2x10^3 ms for one sample image of size 224x224 with a VL-BERT_Large model running on one NVIDIA TITAN RTX GPU of 24 GB memory.
>
> **Time Consumption Analysis:**
> After the region is selected for removal, our proposed coarse-to-fine region removal will be conducted via two main steps: 1) Initial one-pass full region removal/inpainting by SPLp; 2) Triple-pass autoregression region removal/inpainting by SPLf. The triple-pass autoregression strategy ensures fine-grained images and avoids superficial artifacts, as addressed in prior works. After the 1st run of full region inpainting by SPLf, we split the region into M evenly split smaller regions and run the inpainting process for each smaller region, respectively. A similar procedure applies to the third pass of N runs. However, in reality, triple-pass autoregression is a flexible solution as both M and N can be set to any arbitrary positive integers (2<=M<=N) depending on the situation to decrease the overhead time consumption. Hence, The total runs of the inpainting process in the triple-pass strategy is 1 + M + N.
>
> For instance, we conducted the following settings during our study to figure out the trade-off. The base model is a VL-BERT_Large model running on one NVIDIA TITAN RTX GPU of 24 GB. The input image is of size 224x224.
>
> | Exp Index | **# Runs of SPLp** | **# Runs of SPLf (1 + M + N)** | 1st Pass | 2nd Pass | 3rd Pass | Time Consumption (ms) | Accuracy(%) |
> |-----------|:------------------:|:------------------------------:|----------|----------|----------|-----------------------|-------------|
> |           |                    |                                |          | M        | N        |                       |             |
> | 1         | 0                  | 0                              | 0        | 0        | 0        | 10^1                  | 76.07       |
> | 2         | 1                  | 0                              | 0        | 0        | 0        | 10^2                  | 76.18       |
> | 3         | 1                  | 1                              | 1        | 0        | 0        | 2x10^2                | 76.30       |
> | 4         | 1                  | 14                             | 1        | 4        | 9        | 1.3x10^3              | 76.51       |
> | 5         | 1                  | 21                             | 1        | 4        | 16       | 2.2x10^3              | 76.88       |
> | 6         | 1                  | 30                             | 1        | 4        | 25       | 3.5x10^3              | 76.87       |
> | 7         | 1                  | 26                             | 1        | 9        | 16       | 3x10^3                | 76.88       |
>
> Based on the above experiments, we have four observations:
> 1. As in Exp 1, If we do not conduct coarse-to-fine region removal (neither of the two steps will be applied), we essentially apply an occlusion box over the selected region. This is the same approach as the prior works and will cause a relatively low downstream QA accuracy of 76.07.
> 2. Comparing Exp 1-3, we observe that adding the image inpainting process by SPLp and SPLf will both consistently improve the downstream performance.
> 3. Based on Exp 3-5, we can see a noticeable performance improvement trend as we apply more refined region inpainting via increasing the value of M and N.
> 4. In practice, we found that setting M and N to 4 and 16, respectively, generally achieves optimal performance while maintaining reasonable inference time consumption.
>
> We will add these parts of the experiment results and expand our discussion on the experiment section, given an extra page upon acceptance. As mentioned, the triple-pass autoregression strategy is a flexible and general solution. Thus, M and N can be set according to the actual situation to decrease time consumption. We also admit that achieving optimal performance while maintaining minimal time consumption is still a challenging and not fully resolved problem for future work.

---

### Official Review · Reviewer_PvtF · 2023-08-01

**Soundness:** 3

**Excitement:**

3: Ambivalent: It has merits (e.g., it reports state-of-the-art results, the idea is nice), but there are key weaknesses (e.g., it describes incremental work), and it can significantly benefit from another round of revision. However, I won't object to accepting it if my co-reviewers champion it.

**Paper Topic And Main Contributions:**

This paper proposes Adversarial Data Synthesis (ADS) to generate synthetic training and debiased evaluation data and introduce Intra-sample Counterfactual Training (ICT) to assist models in utilizing the synthesized training data.

**Questions For The Authors:**

1. Why you do not select some Sota models, such as ofa or blip2?

**Reasons To Accept:**

This paper analyzed the dataset biases problem in vqa-long, and provide some methods to address the problem.

**Reasons To Reject:**

1. The backbone model is somehow out of date.


**Reproducibility:**

3: Could reproduce the results with some difficulty. The settings of parameters are underspecified or subjectively determined; the training/evaluation data are not widely available.

**Reviewer Confidence:**

2: Willing to defend my evaluation, but it is fairly likely that I missed some details, didn't understand some central points, or can't be sure about the novelty of the work.

---

> ### Author Rebuttal · Authors · 2023-08-29
>
> **SOTA Models:**
> Our method is a generalized data synthesis method that is model-agnostic and can be applied over multiple different base methods. Hence, our objective of this work is not to achieve the SOTA performance on any specific benchmark task but to prove the effectiveness and generalizability of our method. In our initial submission, we selected those high-performing base models specified in the paper as they can be generally applied over VQA-Long benchmarks like VCR, SNLI-VE, etc. The obvious and consistent improvements over multiple high-performing base models have already validated the value of our method.
>
> To further illustrate the effectiveness of our solution, we also conducted additional experiments by applying ADS-ICT over SOTA models like OFA_Large and blip2_flanxxl-t5. We find that OFA_Large can achieve the improved performance of 81.48 with ADS-ICT over the base performance of 80.13 on VQA v2 val and the improved performance of 91.08 with ADS-ICT over the base performance of 90.03 on SNLI-VE dev. Meanwhile, BLIP-2 ViT-G FlanT5 XL can achieve the improved performance of 82.14 with ADS-ICT over the base performance of 81.20 on VQA v2 val dataset. These experiments further verify the advantage of ADS-ICT.
>
> We will provide more details of all the experiment results with SOTA models and expand our discussion on the experiment section, given an extra page upon acceptance.

---

### Official Review · Reviewer_4SCZ · 2023-08-06

**Soundness:** 4

**Excitement:**

4: Strong: This paper deepens the understanding of some phenomenon or lowers the barriers to an existing research direction.

**Paper Topic And Main Contributions:**

The paper proposes a novel method, ADS-ICT, to address biases present in Visual Question Answering tasks with long answer formats (VQA-Long) that consists of two parts, ADS and ICT. ADS comprises two components, ADS-T and ADS-I, which generate synthetic factual and counterfactual text data and images, respectively. ICT utilize a contrastive InfoNCE loss and generated factual and counterfactual samples to train the VQA model. The components together improved the performance on VQA-Long tasks and a new benchmark proposed to evaluate the bias-robustness of models for VQA-Long tasks.

**Questions For The Authors:**

A. what is the specific time overhead for the triple-pass autoregression strategy?

**Reasons To Accept:**

The paper analyzed the biases in the VQA tasks with long answer formats in detail.
The paper proposed ADS-ICT to address the data bias problem with generative adversarial samples and enhanced loss function. These design is validated by extensive experiments for VQA-Long tasks.

**Reasons To Reject:**

The triple-pass autoregression strategy seems to be time-consuming and lead to inference overhead.
The general performance improvement is not obvious against previous method.

**Reproducibility:**

4: Could mostly reproduce the results, but there may be some variation because of sample variance or minor variations in their interpretation of the protocol or method.

**Reviewer Confidence:**

4: Quite sure. I tried to check the important points carefully. It's unlikely, though conceivable, that I missed something that should affect my ratings.

---

> ### Author Rebuttal · Authors · 2023-08-29
>
> **Time Consumption:**
> Our coarse-to-fine region removal method is flexible and generalizable as the number of runs of image inpainting process can be adjusted depending on the scenarios to decrease the time consumption. In practice, we found that setting M and N to 4 and 16, respectively, generally achieves the optimal performance. The time consumption is 2.2x10^3 ms for one sample image of size 224x224 with a VL-BERT_Large model running on one NVIDIA TITAN RTX GPU of 24 GB memory.
>
> **Time Consumption Analysis:**
> After the region is selected for removal, our proposed coarse-to-fine region removal will be conducted via two main steps: 1) Initial one-pass full region removal/inpainting by SPLp; 2) Triple-pass autoregression region removal/inpainting by SPLf. The triple-pass autoregression strategy ensures fine-grained images and avoids superficial artifacts, as addressed in prior works. After the 1st run of full region inpainting by SPLf, we split the region into M evenly split smaller regions and run the inpainting process for each smaller region, respectively. A similar procedure applies to the third pass of N runs. However, in reality, triple-pass autoregression is a flexible solution as both M and N can be set to any arbitrary positive integers (2<=M<=N) depending on the situation to decrease the overhead time consumption. Hence, The total runs of the inpainting process in the triple-pass strategy is 1 + M + N.
>
> For instance, we conducted the following settings during our study to figure out the trade-off. The base model is a VL-BERT_Large model running on one NVIDIA TITAN RTX GPU of 24 GB. The input image is of size 224x224.
>
> | Exp Index | **# Runs of SPLp** | **# Runs of SPLf (1 + M + N)** | 1st Pass | 2nd Pass | 3rd Pass | Time Consumption (ms) | Accuracy(%) |
> |-----------|:------------------:|:------------------------------:|----------|----------|----------|-----------------------|-------------|
> |           |                    |                                |          | M        | N        |                       |             |
> | 1         | 0                  | 0                              | 0        | 0        | 0        | 10^1                  | 76.07       |
> | 2         | 1                  | 0                              | 0        | 0        | 0        | 10^2                  | 76.18       |
> | 3         | 1                  | 1                              | 1        | 0        | 0        | 2x10^2                | 76.30       |
> | 4         | 1                  | 14                             | 1        | 4        | 9        | 1.3x10^3              | 76.51       |
> | 5         | 1                  | 21                             | 1        | 4        | 16       | 2.2x10^3              | 76.88       |
> | 6         | 1                  | 30                             | 1        | 4        | 25       | 3.5x10^3              | 76.87       |
> | 7         | 1                  | 26                             | 1        | 9        | 16       | 3x10^3                | 76.88       |
>
> Based on the above experiments, we have four observations:
> 1. As in Exp 1, If we do not conduct coarse-to-fine region removal (neither of the two steps will be applied), we essentially apply an occlusion box over the selected region. This is the same approach as the prior works and will cause a relatively low downstream QA accuracy of 76.07.
> 2. Comparing Exp 1-3, we observe that adding the image inpainting process by SPLp and SPLf will both consistently improve the downstream performance.
> 3. Based on Exp 3-5, we can see a noticeable performance improvement trend as we apply more refined region inpainting via increasing the value of M and N.
> 4. In practice, we found that setting M and N to 4 and 16, respectively, generally achieves optimal performance while maintaining reasonable inference time consumption.
>
> We will add these parts of the experiment results and expand our discussion on the experiment section, given an extra page upon acceptance. As mentioned, the triple-pass autoregression strategy is a flexible and general solution. Thus, M and N can be set according to the actual situation to decrease time consumption. We also admit that achieving optimal performance while maintaining minimal time consumption is still a challenging and not fully resolved problem for future work.

---

### Meta-Review · Area_Chair_xxhs · 2023-09-18

**Recommendation:** 3

**Metareview:**

This paper focuses on the analysis and mitigation of dataset biases in Visual Question Answering (VQA) tasks with long answer choices. All reviewers appreciated the detailed analysis of bias in this task. But the reviewers also raised concerns on the inference process of the proposed method.

---

### Decision · Program_Chairs · 2023-10-07

**Decision:**

Accept-Findings

**Comment:**

This paper focuses on the analysis and mitigation of dataset biases in Visual Question Answering (VQA) tasks with long answer choices. All reviewers appreciated the detailed analysis of bias in this task. But the reviewers also raised concerns on the inference process of the proposed method.